# Real-World Evidence of Regorafenib Dose Escalation Versus Fixed Dosing in Refractory Metastatic Colorectal Cancer: Results from the ReTrITA Study

**DOI:** 10.3390/cancers17203316

**Published:** 2025-10-14

**Authors:** Carlo Signorelli, Michele Basso, Maria Alessandra Calegari, Annunziato Anghelone, Alessandro Passardi, Chiara Gallio, Alessandro Bittoni, Jessica Lucchetti, Lorenzo Angotti, Emanuela Di Giacomo, Ina Valeria Zurlo, Cristina Morelli, Emanuela Dell’Aquila, Adele Artemi, Donatello Gemma, Alessandra Emiliani, Marta Ribelli, Domenico Cristiano Corsi, Giulia Arrivi, Federica Mazzuca, Federica Zoratto, Marta Schirripa, Francesco Schietroma, Maria Grazia Morandi, Fiorenza Santamaria, Manuela Dettori, Antonella Cosimati, Rosa Saltarelli, Alessandro Minelli, Emanuela Lucci-Cordisco, Mario Giovanni Chilelli

**Affiliations:** 1Medical Oncology Unit, S.Rosa Hospital, ASL Viterbo, 01100 Viterbo, Italyfrancesco.schietroma@asl.vt.it (F.S.); mario.chilelli@asl.vt.it (M.G.C.); 2Oncologia Medica, Comprehensive Cancer Center, Fondazione Policlinico Universitario AgostinoGemelli–IRCCS, 00168 Rome, Italy; 3Department of Medical Oncology, IRCCS Istituto Romagnolo per lo Studio dei Tumori (IRST) “Dino Amadori”, 47014 Meldola, Italy; alessandro.passardi@irst.emr.it (A.P.); chiara.gallio@irst.emr.it (C.G.);; 4Division of Medical Oncology, Policlinico Universitario Campus Bio-Medico, 00128 Rome, Italy; 5Medical Oncology, “Vito Fazzi” Hospital, 73100 Lecce, Italy; inavaleria.zurlo@asl.lecce.it; 6Medical Oncology Unit, Department of Systems Medicine, Tor Vergata University Hospital, 00133 Rome, Italy; 7IRCCS Regina ElenaNational Cancer Institute, 00144 Rome, Italyadele.artemi@uniroma1.it (A.A.); 8Medical Oncology Unit, ASL Frosinone, 03100 Sora, Italy; 9Medical Oncology, Isola Tiberina Hospital, Gemelli Isola, 00186 Rome, Italy; 10Department of Clinical and Molecular Medicine, Oncology Unit, Sant’ Andrea University Hospital, Sapienza University of Rome, 00185 Rome, Italy; 11UOC Oncologia, Ospedale Santa Maria Goretti, ASL Latina, 04100 Latina, Italy; 12Medical Oncology Unit, San Camillo de Lellis Hospital, ASL Rieti, 02100 Rieti, Italy; 13UOC Oncology A, Policlinico Umberto I, 00161 Rome, Italy; fiorenza.santamaria@uniroma1.it; 14Experimental Medicine, Network Oncology and Precision Medicine, Department of Experimental Medicine, Sapienza University of Rome, 00185 Rome, Italy; 15Medical Oncology Department, Ospedale Oncologico Armando Businco, 09121 Cagliari, Italy; manurik@yahoo.it; 16Medical Oncology Department, UO Oncologia Universitaria della Casa della Salute di Aprilia, 04011 Aprilia, Italy; a.cosimati@ausl.latina.it; 17UOC Oncology, San Giovanni Evangelista Hospital, ASL RM5, 00019 Tivoli, Italy; rosa.saltarelli@aslroma5.it; 18Medical Oncology Department, UO Oncologia, Ospedale San Paolo, ASL RM4, 00053 Civitavecchia, Italy; aminelli@hsangiovanni.roma.it; 19UOC Genetica Medica, Dipartimento di Scienze della Vita e Sanità Pubblica, Fondazione Policlinico Universitario Agostino Gemelli, IRCCS, 00168 Rome, Italy; emanuela.luccicordisco@policlinicogemelli.it; 20Medical Oncology Department, Comprehensive Cancer Center, Fondazione Policlinico Universitario Agostino Gemelli, IRCCS, 00168 Rome, Italy

**Keywords:** regorafenib, metastatic colorectal cancer, real-world evidence, dose escalation, fixed dosing, survival outcomes

## Abstract

Regorafenib is an oral multikinase inhibitor approved for patients with metastatic colorectal cancer (mCRC) who have shown progression after standard treatment protocols. The standard initial dose of 160 mg per day often leads to toxicities that may limit treatment adherence. The phase II ReDOS trial indicated that initiating treatment with a lower dose and progressively increasing it enhances tolerability while maintaining efficacy. We performed a comprehensive sub-analysis of the Italian ReTrITA study, involving 729 patients who received treatment with regorafenib. Patients utilising a dose-escalation strategy demonstrated extended progression-free survival and a reduction in severe adverse events relative to those on fixed dosing, with overall survival remaining similar across both groups. The findings demonstrate that flexible dose escalation is a viable method to achieve a balance between efficacy and tolerability in refractory mCRC within a routine practice context, thereby endorsing its application as a practical strategy to enhance treatment outcomes.

## 1. Introduction

Colorectal cancer (CRC) is the fourth most commonly diagnosed malignancy globally and is the second leading cause of cancer-related deaths. The World Health Organisation (WHO) says that, in 2020, more than 1.9 million people were diagnosed with colorectal cancer (CRC) for the first time, and more than 930,000 people died from it around the world [1]. This makes it the second most prevalent type of cancer and the third leading cause of cancer-related deaths. Recent GLOBOCAN forecasts show that the number of people with colorectal cancer around the world will climb by between 63% and 73%, which means that by 2040 there will be 3.2 million more cases and 1.6 million more deaths [2].

Despite notable progress in screening, prevention, and multimodal therapies, metastatic colorectal cancer (mCRC) remains a significant clinical challenge, with around 50–60% of patients ultimately developing distant metastases, the majority of which are unresectable [3,4,5,6]. Although mortality rates among older populations have declined, there has been a paradoxical increase in incidence among younger adults. This highlights the ongoing need for innovative treatment strategies and optimised sequencing methods. These worrying trends show that CRC is becoming more common around the world and that we need to find better and more bearable ways to treat it, especially for those whose cancer has spread and is resistant to treatment.

In salvage scenarios, therapeutic options are limited [7,8,9,10,11,12,13,14,15,16,17,18,19,20,21,22]. Regorafenib, an oral multikinase inhibitor, received FDA approval in 2012 for the treatment of patients with mCRC previously treated with standard chemotherapy and targeted therapies. Subsequent approvals extended its indication to gastrointestinal stromal tumours (GIST) and hepatocellular carcinoma (HCC), confirming its role as an established therapeutic option in refractory solid tumours [23,24]. It has been recognised as a standard treatment for refractory mCRC, supported by key phase III trials. The CORRECT and CONCUR studies indicated a survival benefit of regorafenib compared to placebo, accompanied by manageable yet clinically significant toxicities such as hand–foot skin reaction, fatigue, hypertension, and hepatotoxicity [25,26]. Investigations in real-world settings, including CONSIGN, REBECCA, and CORRELATE, have substantiated the safety and efficacy of regorafenib across various populations [27,28,29]. The phase II ReDOS trial implemented a dose-escalation approach, demonstrating that gradual titration from 80 mg to 160 mg enhanced treatment adherence and tolerability relative to the standard 160 mg starting dose, thus supporting alternative dosing strategies [30].

The ReTrITA (Regorafenib and Trifluridine/Tipiracil in Italian Patients) study was developed as one of the largest real-world, multicentre retrospective analyses in Italy, building on this background. Between 2012 and 2023, 1156 patients with refractory metastatic colorectal cancer (mCRC) were enrolled across 17 oncology centres and treated with regorafenib and/or trifluridine/tipiracil, either as monotherapy or in sequential regimens. The study demonstrated comparable survival outcomes for the two drugs when used separately; however, it indicated significantly enhanced overall and progression-free survival when regorafenib was administered before trifluridine/tipiracil (R/T) compared to the reverse order (T/R). The R/T sequence resulted in a median overall survival (OS) of 16.6 months, compared to 12.6 months for T/R, and a median progression-free survival (PFS) of 11.5 months versus 8.5 months, respectively. The results were consistent across multiple clinically relevant subgroups, underscoring the significance of treatment sequence in the management of late-line mCRC [31].

The ReTrITA study adhered to the Declaration of Helsinki and obtained approval from the Ethics Committee of Area 4 Lazio, Rome, Italy (protocol number 29-2024; approval date 4 March 2024). Patient data were anonymised, and informed consent was exempted in compliance with national data protection regulations.

This sub-analysis examines patients treated with regorafenib in the ReTrITA cohort, comparing outcomes between those managed with the ReDOS escalation strategy and those receiving standard or modified non-ReDOS schedules. This study aims to elucidate the potential benefits of dose optimisation in routine clinical practice by integrating efficacy, safety, and subgroup analyses, thereby enhancing the evidence base for personalised regorafenib therapy in refractory mCRC. Figure 1 displays the study design.

## 2. Patients and Methods

### 2.1. Study Overview

The ReTrITA study was published on 18 June 2025. This retrospective, multicentre analysis encompassed 1156 patients with refractory metastatic colorectal cancer who were treated from 2012 to 2023 at 17 Italian cancer centres. Patients were assigned to receive either trifluridine/tipiracil (T) or regorafenib (R) as monotherapies, or in sequential treatments: T/R or R/T. The inclusion criteria included adult patients with metastatic colorectal cancer who were refractory to standard therapies and received treatment with regorafenib and/or trifluridine/tipiracil between 2012 and 2023 at participating centres [31].

Key findings indicated no significant differences in overall survival (OS) or progression-free survival (PFS) between R and T when administered as monotherapies (e.g., OS: ~5.0 vs. 5.9 months, PFS: ~3.2 vs. 3.3 months; non-significant). The R/T sequence demonstrated a significant enhancement in outcomes relative to T/R, with a median overall survival (OS) of 16.6 months compared to 12.6 months (*p* = 0.0004) and a median progression-free survival (PFS) of 11.5 months versus 8.5 months (*p* < 0.0001). The survival benefit was consistent across clinically relevant subgroups, including older patients, individuals with RAS mutations, and those with microsatellite-stable tumours. Toxicity profiles were consistent with expectations: haematologic toxicities were predominant with T, while non-hematologic toxicities were more common with R. The R/T sequence was linked to a higher incidence of grade 3/4 toxicities, yet exhibited a lower occurrence of neutropenia [31].

### 2.2. Outcome Parameters

This sub-analysis of the ReTrITA study concentrated on overall survival (OS) and progression-free survival (PFS) as the main endpoints. Overall survival (OS) is defined as the time from the initiation of regorafenib until death from any cause, whereas progression-free survival (PFS) is defined as the duration from the start of treatment to the first occurrence of clinically or/and radiologically confirmed disease progression or death. Secondary endpoints comprised the objective response rate (ORR), which is the percentage of patients achieving a complete response (CR) or partial response (PR), and the disease control rate (DCR), defined as CR, PR, or stable disease (SD), evaluated in accordance with RECIST v1.1. Furthermore, duration of response (DoR) was defined as the interval from the initiation of regorafenib treatment to disease progression or death in patients who attained complete response (CR) or partial response (PR). Safety outcomes comprised the incidence of treatment-related grade 3/4 adverse events (AEs), evaluated in accordance with NCI CTCAE v4.0 criteria.

### 2.3. Drug Administration

In this sub-analysis we focused exclusively on patients administered regorafenib, as illustrated in Figure 1 of the original study design. Regorafenib was taken orally once daily for 21 days within each 28-day cycle. The no-ReDOS group received an initial dose of 160 mg/day, with subsequent modifications determined by the clinician’s judgement. The ReDOS group executed a dose-escalation protocol, initiating with 80 mg/day in week 1, advancing to 120 mg/day in week 2, and reaching 160 mg/day in week 3, contingent upon tolerance, and thereafter remaining at the maximum tolerated dose of up to 160 mg/day. Both groups permitted dose reductions, interruptions, or delays contingent upon patient tolerance. Treatment persisted until disease progression, intolerable toxicity, decline in performance status, or withdrawal of consent occurred.

### 2.4. Statistical Analysis

Analyses were conducted utilising MedCalc v19.4 (MedCalc Software, Ostend, Belgium). Descriptive statistics provided a summary of demographic, clinical, and treatment characteristics. Overall survival (OS) and progression-free survival (PFS) were estimated using Kaplan–Meier methods, and survival comparisons between the ReDOS and non-ReDOS groups were conducted using the log-rank test. Hazard ratios (HRs) and 95% confidence intervals (CIs) were calculated employing Cox proportional hazards models. Comparisons of categorical variables, including adverse event incidence, utilised Fisher’s exact test or chi-square tests as deemed appropriate. DoR was presented with median values and ranges. A two-sided *p*-value of ≤0.05 was deemed statistically significant.

Exploratory subgroup analyses were performed across clinically relevant categories, including age, sex, ECOG performance status, RAS mutation status, MMR status, primary tumour location (right colon, left colon, rectum), prior exposure to anti-EGFR/VEGF agents, history of adjuvant treatment, use of rechallenge strategies, and metastatic disease pattern. Forest plots presented hazard ratios and 95% confidence intervals for OS and PFS across these subgroups to facilitate interpretation. All subgroup analyses were exploratory, and *p*-values were not adjusted for multiplicity; thus, these findings should be interpreted cautiously.

The analysis incorporated all patients administered regorafenib to reduce selection bias. The lead investigator, who also served as the data manager, had complete access to the database and conducted statistical analyses. Patient selection was performed by independent investigators who were blinded to clinical outcomes. All outcomes and definitions were established prior to the collection of data. However, due to the retrospective design, the results should be regarded as exploratory.

### 2.5. Ethical Approval

The ReTrITA study adhered to the Declaration of Helsinki and received approval from the Ethics Committee of Area 4 Lazio, Rome, Italy (protocol number 29-2024, approved on 4 March 2024). Patient data were anonymised to maintain confidentiality, and informed consent was waived in instances where patients were unreachable or declined to provide it, in compliance with Italian data protection regulations.

## 3. Results

### 3.1. Patient Characteristics

Table 1 summarises the baseline demographic and clinical characteristics of patients involved in this ReTrITA sub-study, we used bold for statistically significant numbers. This sub-analysis included 713 patients who were treated with regorafenib. Among them, 313 (43.9%) were treated following the ReDOS escalation strategy, whereas 400 (56.1%) were managed with no-ReDOS schedules (*p* = 0.0011).

The median age was similar across groups (58 vs. 59 years, *p* = 0.9263), and the distribution of patients aged ≥ 70 years was also comparable (43.7% vs. 37.8%, *p* = 0.1042). The sex distribution was balanced, with males constituting 58.2% and females 60.8%, *p* = 0.4824. The prevalence of RAS mutations was similar across cohorts (56.0% vs. 58.8%, *p* = 0.6549). The distribution of primary tumour locations was similar; however, a higher percentage of right-sided primaries was observed in the no-ReDOS group (35.3% vs. 30.3%, *p* = 0.1948).

Significant variations were observed in mismatch repair (MMR) status. In the ReDOS group, a higher prevalence of pMMR disease was observed (78.0% compared to 37.8%). Conversely, the no-ReDOS cohort exhibited a significant proportion of missing or untested MMR status (60.5% versus 18.5%, *p* < 0.0001).

A notable difference in performance status was observed, with a higher percentage of ECOG 0 patients in the ReDOS cohort (37.4% vs. 28.3%) and a larger proportion of ECOG 2 patients in the no-ReDOS group (12.8% vs. 6.1%, *p* = 0.0017). Prior adjuvant therapy occurred more frequently in ReDOS patients (30.0% compared to 24.3%), but this difference was not statistically significant (*p* = 0.0838).

Liver-only involvement in metastatic disease was more prevalent in the ReDOS group (17.6% vs. 9.0%), while the no-ReDOS group showed a higher incidence of combined liver and extrahepatic disease (56.8% vs. 48.2%, *p* = 0.0019).

Disparities in chemotherapy administration were also noted. The ReDOS group consistently received intensive first-line and second-line regimens, including doublets or triplets, with minimal data missing. In contrast, the no-ReDOS group exhibited a significant percentage of patients with unknown treatment history (31.0% in first line; 35.3% in second line). Both differences exhibited statistical significance (*p* < 0.0001).

Exposure to biological agents demonstrated considerable variability. In first-line treatment, anti-EGFR therapy was more prevalent in the ReDOS cohort (31.6% vs. 20.0%), while a higher proportion of no-ReDOS patients had not received any targeted agent (40.0% vs. 17.2%; *p* < 0.0001). In the second-line setting, the utilisation of anti-VEGF was markedly higher in ReDOS patients (66.1% compared to 45.0%). Conversely, no-ReDOS patients exhibited a higher prevalence of no exposure to biological agents (51.3% versus 26.8%; *p* < 0.0001).

These findings indicate that while baseline demographic factors (age, sex, RAS status, tumour location) were largely comparable, significant imbalances were observed in ECOG performance status, metastatic distribution, MMR testing, prior chemotherapy intensity, and the use of biological agents. These differences must be taken into account when analysing survival and toxicity outcomes, as they may indicate variations in clinical practice, patient selection, and data completeness between the two treatment groups.

### 3.2. Survival Outcomes

The sub-analysis indicated that overall survival (OS) was not significantly different between the two groups. The median overall survival (OS) was 7.4 months (95% confidence interval [CI], 6.4–8.9) for the ReDOS group and 6.7 months (95% CI, 6.0–10.1) for the no-ReDOS cohort, with mean survival times being nearly equivalent at 14.2 months versus 14.3 months. The log-rank test indicated no statistically significant difference (χ^2^ = 0.008, DF = 1, *p* = 0.9298; HR 1.00, 95% CI 0.85–1.18) (Figure 2). The ReDOS escalation strategy significantly prolonged progression-free survival (PFS). The median progression-free survival (PFS) was 3.1 months (95% confidence interval [CI], 3.0–35.6) in the ReDOS group, in contrast to 3.9 months (95% CI, 3.5–66.2) in the no-ReDOS group. The log-rank test indicated a significant benefit for the ReDOS cohort (χ^2^ = 11.55, DF = 1, *p* = 0.0007; HR 0.76, 95% CI 0.65–0.89) (Figure 3).

These findings suggest that while overall survival did not significantly differ between dosing strategies, progression-free survival improved in patients on the ReDOS escalation schedule, highlighting a potential benefit of gradual dose titration in delaying disease progression. The difference in overall survival and progression-free survival outcomes may indicate the influence of subsequent therapies, patient selection, or the retrospective nature of the study design. The statistically significant improvement in progression-free survival supports the hypothesis that the ReDOS strategy could help with early disease control while achieving overall survival rates similar to those of no-ReDOS regimens.

### 3.3. Summary of Efficacy Outcomes

The efficacy results of this study, as detailed in the preceding sections, are summarised in Table 2. The objective response rates (ORR) were low in both cohorts, with 2.0% for ReDOS and 2.6% for no-ReDOS (*p* = 0.4386). The disease control rate (DCR) was comparable, recorded at 23.2% and 25.3%, respectively (*p* = 0.6580). The duration of response (DoR) was greater in patients not receiving ReDOS (15.4 months compared to 8.9 months), though this difference did not achieve statistical significance (HR 1.68, 95% CI 0.35–7.94; *p* = 0.5944). The findings indicate similar overall survival across strategies, with the no-ReDOS group demonstrating a progression-free survival advantage, although response rates and disease control were constrained in both cohorts.

### 3.4. Comparative Analysis of Subgroups Survival

The comparative evaluation of regorafenib dose-escalation (ReDOS) versus fixed-dose (no-ReDOS) strategies across clinically relevant subgroups provides important insights into patient outcomes in refractory metastatic colorectal cancer.

#### 3.4.1. Overall Survival (OS)

In the overall population, OS did not differ significantly between ReDOS and no-ReDOS, with hazard ratios (HRs) close to unity and confidence intervals crossing 1. Subgroup analyses confirmed this general finding, although selected signals emerged. Among patients aged ≥ 70 years, OS was 7.1 months under ReDOS compared with 6.6 months for no-ReDOS (HR 0.89, *p* = 0.378). In contrast, younger patients showed nearly identical outcomes (7.9 vs. 7.1 months; HR 0.91, *p* = 0.435). ECOG PS stratification revealed that patients with PS2 experienced worse survival with ReDOS (3.7 vs. 4.4 months; HR 1.74, *p* = 0.046), whereas outcomes were comparable in PS = 0–1.

Molecular subgroups, such as RAS wild-type and mutant, showed no significant differences, and frontline exposure to anti-EGFR or anti-VEGF did not alter OS outcomes. Patients with dMMR exhibited a longer OS with ReDOS (13.1 months compared to 10.0 months), although the results did not achieve statistical significance. Tumour sidedness suggested a trend: right-sided primaries had modestly better outcomes under ReDOS (8.1 vs. 6.1 months; HR 0.84), while rectal tumours favoured no-ReDOS (8.2 vs. 10.0 months; HR 1.38) (Figure 4).

Overall, Chi-squared analyses supported the absence of significant OS differences across most subgroups, with the exception of ECOG PS2, where ReDOS was inferior.

#### 3.4.2. Progression-Free Survival (PFS)

In contrast, subgroup analyses of PFS always showed that the no-ReDOS strategy was better. For people 70 years and older, the median PFS was 3.1 months with ReDOS and 3.7 months without it (HR 1.46, *p* = 0.0031). A comparable tendency was noted in younger patients (<70 years), where no-ReDOS resulted in an extended PFS (3.9 vs. 3.0 months; HR 1.22, *p* = 0.057). Performance level was a crucial factor: PS1 patients experienced 3.9 months without ReDOS compared to 3.1 months with ReDOS (HR 1.37, *p* = 0.0029), and PS2 patients also exhibited poorer outcomes with ReDOS (2.6 vs. 2.8 months; HR 1.83, *p* = 0.027) (Figure 5).

These results were even stronger when genetic and clinical factors were used to group the data. Patients with RAS wild-type and mutant genotypes both experienced enhanced benefits from no-ReDOS, with hazard ratios of 1.49 and 1.23, respectively. Patients with pMMR also had longer PFS with no-ReDOS (3.7 months vs. 3.1 months; HR 1.36, *p* = 0.0036). Tumour location studies corroborated suboptimal outcomes with ReDOS in right-sided primary (HR 1.41, *p* = 0.020) and rectal tumours (HR 1.74, *p* = 0.001).

Further studies indicated that male patients and individuals with previous anti-EGFR exposure exhibited considerably enhanced disease management under no-ReDOS, but prior anti-VEGF medication did not significantly affect the comparison. Both the adjuvant and non-adjuvant groups had longer PFS with no-ReDOS, but only the non-adjuvant group reached significance (HR 1.36, *p* = 0.001). Global Chi-squared analysis corroborated considerable heterogeneity across ECOG PS (χ^2^ = 47.47, df = 5, *p* < 0.0001), metastatic site (χ^2^ = 33.66, df = 5, *p* < 0.0001), tumour location (χ^2^ = 18.85, df = 5, *p* = 0.0021), and adjuvant exposure (χ^2^ = 19.53, df = 3, *p* = 0.0002).

### 3.5. Survival Outcomes by Initial Regorafenib Dosing Strategy

In a supplementary analysis, patients were categorised into three groups based on initial regorafenib dosing: ReDOS escalation (*n* = 313), fixed reduced dose < 160 mg (*n* = 142), and standard 160 mg dose (*n* = 258). The median overall survival (OS) was 7.4 months (95% CI, 6.4–8.9) for the ReDOS group, 8.3 months (95% CI, 6.3–10.0) for the <160 mg group, and 6.1 months (95% CI, 6.0–10.1) for the 160 mg group, with no statistically significant differences observed among the groups (log-rank χ^2^ = 1.95, *p* = 0.3781). Pairwise hazard ratios indicated no overall survival benefit: HR 0.90 (95% CI, 0.72–1.11) for ReDOS compared to <160 mg and HR 1.05 (95% CI, 0.87–1.25) for ReDOS compared to 160 mg (Figure 6A).

Progression-free survival demonstrated significant differences among dosing groups (log-rank χ^2^ = 11.77, *p* = 0.0028). The median PFS was 3.1 months (95% confidence interval [CI], 3.0–35.6) for ReDOS, 3.8 months (95% CI, 3.4–28.1) for doses under 160 mg, and 3.9 months (95% CI, 3.5–66.2) for 160 mg doses. Hazard ratios indicated a decreased risk of progression or mortality with ReDOS compared to <160 mg (HR 0.83, 95% CI, 0.67–1.02) and 160 mg (HR 0.75, 95% CI, 0.63–0.89) (Figure 6B).

The findings indicate that while OS was not influenced by the initial dose, PFS outcomes were enhanced by the ReDOS escalation strategy, implying that gradual dose titration could enhance early disease management without negatively impacting overall survival.

## 4. Safety

This subanalysis of the ReTrITA study examined real-world regorafenib administration patterns in metastatic colorectal cancer, systematically comparing the incidence of treatment-related grade 3/4 adverse events (AEs) between patients receiving regorafenib via the ReDOS escalation strategy and those treated with no-ReDOS regimens (Table 3).

Overall, grade 3/4 adverse events were found in both groups, yet their distribution exhibited significant differences. The total number of grade 3/4 events was similar (141 in the ReDOS group compared to 203 in the no-ReDOS group, *p* = 0.0008). However, the proportion of patients experiencing at least one grade 3/4 toxicity was significantly lower in the ReDOS cohort (35.4% versus 39.5%, *p* = 0.0042). This indicates that stepwise dose escalation may enhance tolerability, even with comparable overall event counts.

In the analysis of toxicities, haematologic grade 3/4 events were infrequent and showed no significant difference between the groups (7.8% vs. 10.3%, *p* = 0.0771). Non-hematologic toxicities predominated the safety profile, constituting more than 90% of high-grade events in both groups. The relative frequency was notably elevated in the ReDOS group (92.2% versus 89.7%, *p* = 0.0032), indicating a shift in the toxicity profile associated with the dosing method used.

Anaemia emerged as the predominant grade 3/4 haematologic toxicity, affecting 63.6% of patients in the ReDOS group and 61.9% in the no-ReDOS group. The occurrence of neutropenia, febrile neutropenia, and thrombocytopenia was low and uniformly distributed across cohorts, underscoring the limited haematologic effects of regorafenib in this setting.

Non-hematologic toxicities displayed increased variability. Fatigue was the most common severe adverse event, affecting 39.0% of patients with Grade 3/4 toxicities in the ReDOS group, in contrast to 23.6% in the no-ReDOS group. The hand-foot skin reaction was notable, with a higher incidence in no-ReDOS patients (21.2% versus 16.9%). Hypertension and liver dysfunctions were observed at similar rates among cohorts, whereas diarrhoea and skin disorders were more prevalent in the no-ReDOS group. The “other” category, which encompassed various less frequent toxicities, was also more represented in the no-ReDOS cohort (27.1% vs. 20.3%).

The findings suggest that regorafenib toxicity in the ReTrITA study aligned with the known safety profile of the drug; however, the dosing strategy affected the pattern and distribution of adverse events. The ReDOS escalation approach correlated with a reduced incidence of severe toxicities, especially regarding cumulative burden; however, specific non-hematologic toxicities, including fatigue, persisted as significant concerns. This suggests that personalised dose optimisation strategies could improve the tolerability of regorafenib while maintaining its overall safety profile.

## 5. Discussion

This sub-analysis of the ReTrITA study presents a substantial real-world dataset concentrated on regorafenib dosing strategies in refractory metastatic colorectal cancer (mCRC). The study compares outcomes of patients treated with a ReDOS-like dose escalation against standard or modified fixed dosing (no-ReDOS), providing insights into how titration strategies can optimise the balance between efficacy and tolerability in clinical practice. The findings support the concept that the efficacy of regorafenib can be maintained and, in certain instances, enhanced when the drug is administered via a flexible escalation strategy that emphasises tolerability and continuity of treatment.

In the context of previous real-world observational studies (Table 4), notable similarities and differences are observed. The current ReTrITA sub-analysis, focussing on regorafenib-treated patients (*n* = 729), indicated a median overall survival (OS) of 7.8 to 8.9 months across groups, with progression-free survival (PFS) ranging from 2.8 to 4.0 months, demonstrating favourable outcomes compared to other observational cohorts.

By contrast, the Spanish RE-SEARCH study (*n* = 242, 15 centres) primarily investigated real-world dosing patterns [32]. Most patients (63.8%) initiated regorafenib at <160 mg, and dose-escalation from reduced starting doses became increasingly frequent over time, especially after 2019. The median PFS was 3.0 months, with no significant differences in efficacy observed across dose patterns. Patients initiating treatment at 160 mg, followed by dose reductions, exhibited the highest incidence of grade 3–4 toxicities, recorded at 40.6%. ReTrITA demonstrates the clinical advantages of dose-escalation regarding PFS and tolerability, whereas RE-SEARCH illustrates the gradual adoption of reduced starting doses followed by escalation by Spanish oncologists to enhance treatment feasibility. Both studies support the idea that flexible dosing strategies can preserve efficacy and improve safety in the routine management of refractory metastatic colorectal cancer.

Nakashima et al. (*n* = 2530) assessed dose appropriateness in Japan, considering body weight as a factor. Lighter patients demonstrated tolerance to low-dose regorafenib (≤120 mg), achieving outcomes similar to heavier patients on full doses, with overall survival (OS) ranging from 7.5 to 8.0 months across strata [33]. The results of ReTrITA corroborate this evidence, indicating that increasing the dosage from 80 mg does not negatively impact survival rates but rather improves tolerability, thereby underscoring the significance of personalised treatment approaches.

Yan et al. conducted an analysis of 77 patients in China, reporting an overall survival (OS) of 17.8 months and progression-free survival (PFS) of 4.6 months, which are longer than those observed in most Western cohorts. The inclusion of patients in earlier treatment lines and those receiving combinations with immunotherapy impacts the comparability of these findings [34]. However, their observation that intermediate doses (120 mg) resulted in the longest overall survival aligns with the ReTrITA finding that escalation strategies, as opposed to a standard 160 mg initiation, may provide the most favourable therapeutic index.

When compared with the large US real-world retrospective analysis by Bekaii-Saab et al. (703 patients), notable similarities and differences emerge. In the US cohort, the adoption of flexible dosing increased markedly after inclusion of the ReDOS strategy in the NCCN Guidelines (from 21% to 45%), with a corresponding improvement in treatment persistence, as reflected by a higher proportion of patients reaching the third cycle of therapy (36.5% pre-ReDOS vs. 45.5% post-ReDOS). Although survival outcomes were not the primary focus of that study, these data support the feasibility and increasing uptake of dose individualization in everyday practice [35]. Compared with the ReTrITA findings, the US study provides complementary evidence that dose escalation is being translated into clinical benefit through longer treatment duration.

The CORRELATE Taiwan analysis (128 patients) offers additional insights from an Asian setting. In this population, 71.9% of patients started regorafenib at a dose lower than 160 mg/day, and the median OS was 11.6 months with a PFS of 2.2 months [36]. This OS is numerically longer than reported in both the global CORRELATE population and in our ReTrITA sub-analysis, although cross-study comparisons should be made with caution due to potential differences in baseline characteristics, mutation prevalence, and subsequent therapies. Of note, the Taiwanese patients demonstrated a higher prevalence of RAS and BRAF mutations compared to other populations, which may have influenced outcomes.

In elderly cohorts, such as the Japanese multicentre study (Hatori et al.), reduced-dose regorafenib demonstrated sustained efficacy while reducing toxicity, with overall survival (OS) around 7.0 months [37]. This finding aligns with the ReTrITA subgroup aged ≥ 70 years, where dose escalation resulted in a median OS nearing 8.0 months.

The analysis by Peeters et al. (2024) [38], conducted in France, Italy, and Belgium with a sample of 355 patients, primarily examined treatment persistence rather than survival endpoints. Patients were categorised into three groups: ReDOS-like initiation at <160 mg/day, dose-adjusted initiation at 160 mg/day with subsequent reductions, and standard fixed-dose at 160 mg/day. The median duration of treatment (DoT) was longer in the dose-adjusted group (1.9 months) compared with both the ReDOS-like (1.4 months) and standard (1.0 months) groups. Moreover, the proportion of patients reaching at least three cycles of therapy was substantially higher in the flexible dosing groups (93% in dose-adjusted, 67% in ReDOS-like) compared with 64% in the standard arm. Patients in the flexible groups exhibited poorer baseline prognostic factors, such as ECOG PS ≥ 2, multiple metastatic sites, and greater rates of RAS mutations, yet they achieved longer treatment persistence [38]. ReTrITA and Peeters offer complementary insights when considered collectively. ReTrITA establishes that dose escalation improves disease control without compromising overall survival. Peeters emphasises that flexible regimens, including escalation or dose adjustment, enhance treatment adherence and persistence, even in patients with less favourable characteristics. Both studies converge on the conclusion that rigid adherence to the 160 mg daily starting dose is rarely feasible in clinical practice and that tailoring regorafenib dosing represents a pragmatic strategy to optimise both tolerability and continuity of care in refractory metastatic colorectal cancer.

Collectively, these real-world studies all point toward the same conclusion: regorafenib is rarely administered in real life at the rigid 160 mg starting dose used in pivotal trials. Instead, physicians routinely adapt the dose according to patient characteristics, tolerability, and guidelines. While outcomes vary slightly across regions, the consistency of PFS (~3 months) underscores the reproducibility of regorafenib activity when managed flexibly.

From a clinical perspective, this real-world sub-analysis illustrates the significance of administering the appropriate dosage of regorafenib to patients with refractory mCRC. The ReDOS experiment was the first to test the method of dosage escalation. Then, the ReTrITA group showed that flexible titration works in the real world and might help patients take the drug as directed without making it less effective. This way, doctors can change a patient’s treatment based on how well they are doing overall and how well it is working for them. In the late-line setting, where there are not many therapy alternatives and the goal is to remain up with life, this is really important. The findings corroborate recent observational studies indicating that adaptive dosing effectively balances efficacy and tolerability. Patients undergoing progressive escalation may experience prolonged exposure to the therapy, yielding cumulative benefits over time due to their increased adherence and reduced likelihood of discontinuation. The ReTrITA results support the hypothesis that regorafenib can be a helpful aspect of tailored treatment programmes.

Taken together, the strengths of the ReTrITA sub-analysis include a large, homogeneous cohort of regorafenib-treated patients, a systematic comparison between escalation and fixed dosing, and detailed subgroup analyses with forest plots that confirm the consistency of progression-free survival benefits across clinically relevant categories. It is essential to recognise the limitations. The retrospective design presents potential biases, such as non-standardised dose assignment and variability in clinical practice. Baseline imbalances were significant: variations in ECOG performance status, metastatic patterns, the extent of missing or untested MMR, prior chemotherapy intensity, and prior biologic exposure—especially the greater proportion of unknown histories in no-ReDOS—may affect both tolerability and outcomes. The presence of missing data and variability in prior treatments among different centres restricts comparability. Furthermore, although the advantages in PFS were statistically significant, the differences in OS were modest, indicating both the refractory characteristics of the population and the confounding influence of subsequent therapies. All subgroup signals should be regarded as generating hypotheses. In contrast to certain Asian studies, pharmacogenomic and body-weight-adjusted analyses were not accessible. The observational, retrospective nature of this analysis, combined with non-random allocation to dosing strategies, introduces the potential for confounding by indication.

It is important to recall that the groups were different at the outset of the study, especially when it came to ECOG performance status, MMR profile, and previous use of biologic drugs. These variations may have made the treatment less effective. These differences show how unpredictable real-world clinical practice may be. This is because treatment decisions are based on things like the patient’s health, the biology of the disease, and past treatments, not strict study inclusion requirements. So, even though there is always a chance of confounding, the general direction and size of the connections we identified support the conclusions, as we used a non-randomised, retrospective strategy.

Subgroup survival analysis reveals significant variations in overall survival and progression-free survival patterns. Excluding individuals with ECOG PS2, who exhibited suboptimal performance with ReDOS, overall survival outcomes were generally similar across all dosing regimens. Trials assessing progression-free survival consistently indicated that the fixed-dose strategy outperformed the alternative approach across nearly all clinically significant categories. This discrepancy indicates that the method of drug administration may not significantly influence survival; however, PFS suggests that in specific patient populations, a fixed dose could improve disease management.

The data demonstrate that no-ReDOS dosage consistently improves PFS, especially in patients with a history of anti-EGFR therapy, older adults, individuals with right-sided or rectal primaries, or RAS wild-type tumours. Personalised treatment approaches are essential, as ReDOS may be ineffective and potentially harmful in frailer or high-risk subgroups.

The dose-escalation strategy worked in real life and had a solid safety record. It also made patients more likely to persist with therapy without diminishing their odds of surviving. The fixed-dose group had a longer DoR than the dose-escalation group, which is interesting because the dose-escalation group tended to have higher PFS. There are several possible reasons for this. These include bias in deciding on patients in the real world, where healthier patients may have been more likely to be able to handle starting with a fixed dose; an imbalance in the number of patients in each group; and possible differences in how drugs work or how they move through the body that could change the timing and consolidation of tumour response. These findings necessitate further examination in subsequent research that include pharmacokinetic monitoring and a standardised method for assessing responsiveness.

In aggregate, ReTrITA endorses a pragmatic escalation strategy that focusses on optimising early tolerability to maintain treatment and enhance PFS, while acknowledging that OS in refractory metastatic colorectal cancer (mCRC) is significantly affected by subsequent therapies and patient frailty. The conclusions align with randomised evidence and current real-world practices that implement flexible dosing. Prospective studies ought to (i) validate escalation compared to fixed starts with standardised toxicity management, (ii) incorporate exposure metrics such as early cumulative dose and metabolite levels, and (iii) refine the selection of individuals for whom escalation is most beneficial (e.g., older adults, ECOG 1–2), ideally within a learning health system framework.

## 6. Conclusions

This analysis demonstrates that regorafenib dose escalation is a feasible and well-tolerated strategy for patients with refractory mCRC, enhancing treatment persistence and optimising patient management while maintaining overall survival rates. The findings endorse individualised dosing as an effective approach to improve tolerability and sustain clinical benefit in standard practice. In comparison to other extensive real-world studies, including CORRELATE, RE-SEARCH, and the Japanese and U.S. cohorts, our findings reinforce the emerging consensus that individualised dosing, as opposed to standard full-dose initiation, optimises both efficacy and tolerability. The strengths of this study include a substantial Italian multicentre dataset, thorough subgroup analyses, and relevance to real-world applications. The retrospective design and lack of prospective validation highlight the necessity for confirmatory trials. In the future, the integration of escalation protocols with the biomarker-driven selection and optimisation of supportive care may enhance regorafenib’s role in the continuum of care for mCRC.

## Figures and Tables

**Figure 1 cancers-17-03316-f001:**
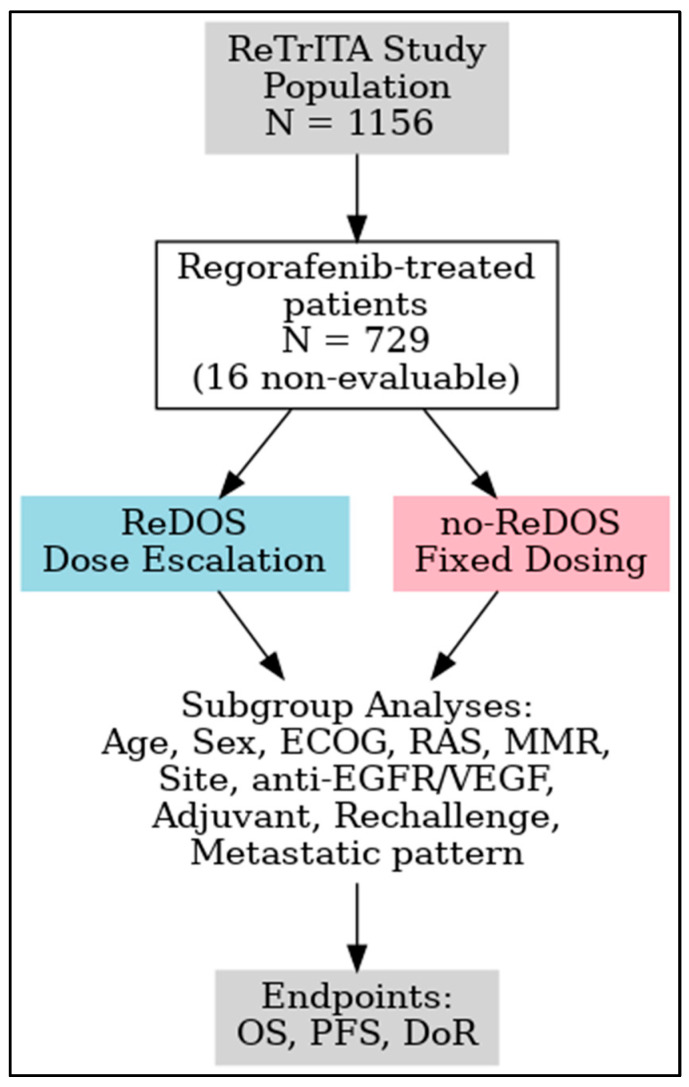
Study design of the ReTrITA sub-analysis. Abbreviations: T, trifluridine/tipiracil; R, regorafenib.

**Figure 2 cancers-17-03316-f002:**
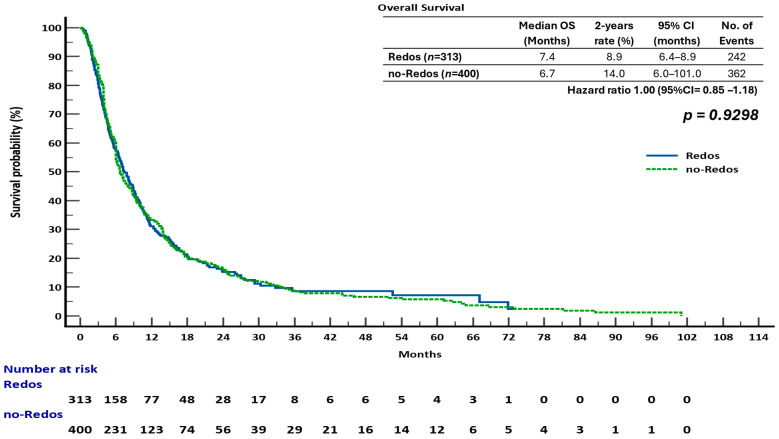
Overall survival (OS) in patients treated with regorafenib according to dosing strategy.

**Figure 3 cancers-17-03316-f003:**
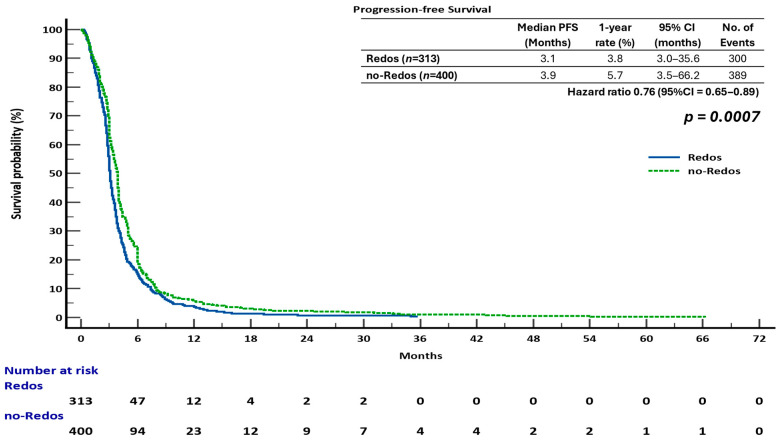
Progression-free survival (PFS) in patients treated with regorafenib according to dosing strategy.

**Figure 4 cancers-17-03316-f004:**
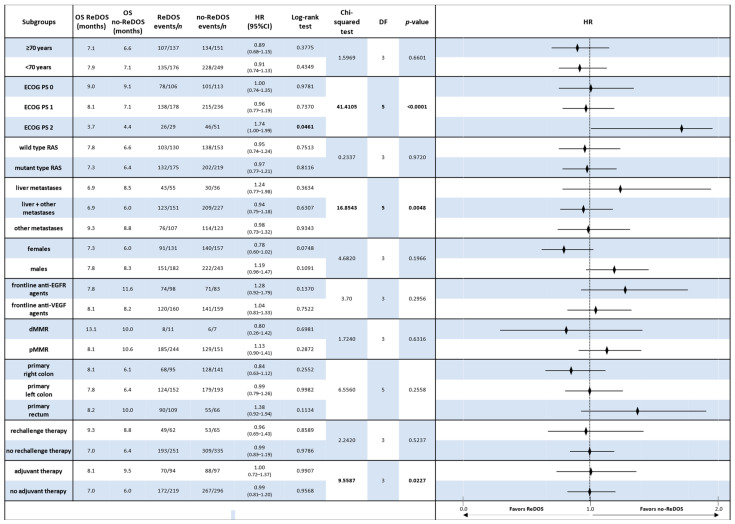
Forest plot of overall survival (OS) by clinically relevant subgroups, comparing patients treated with regorafenib according to the ReDOS escalation strategy versus no-ReDOS dosing.

**Figure 5 cancers-17-03316-f005:**
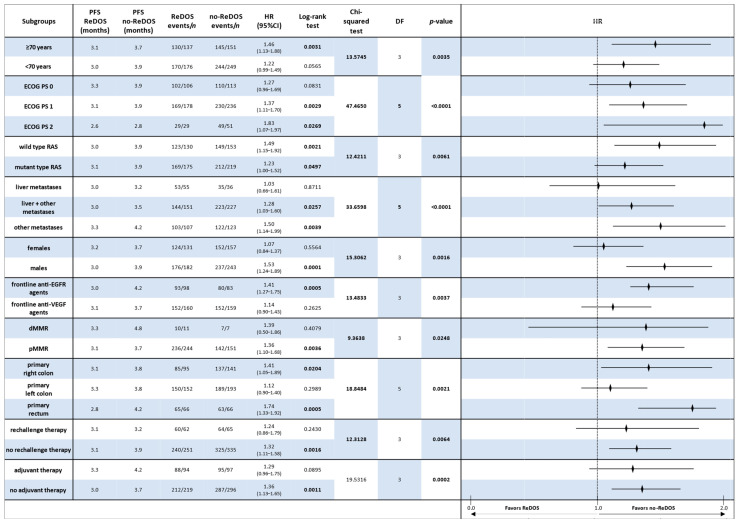
Forest plot of progression-free survival (PFS) by clinically relevant subgroups, comparing patients treated with regorafenib according to the ReDOS escalation strategy versus no-ReDOS dosing.

**Figure 6 cancers-17-03316-f006:**
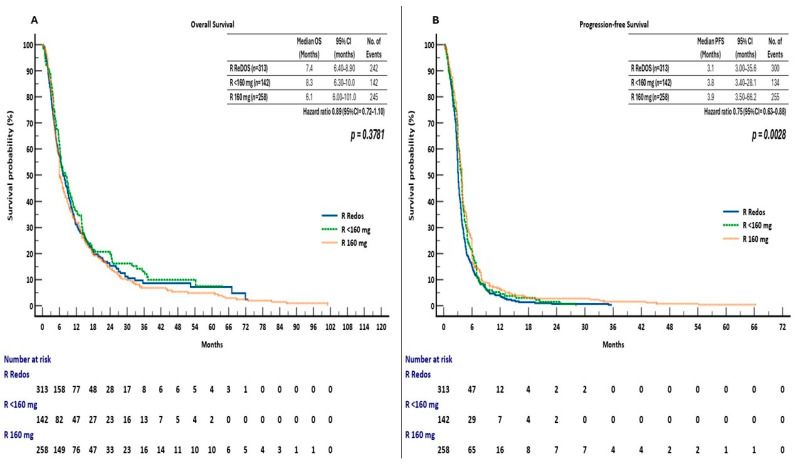
(**A**) Overall survival (OS) and (**B**) Progression-free survival (PFS) by initial regorafenib dosing strategy in the ReTrITA sub-analysis (ReDOS escalation; fixed reduced dose < 160 mg; standard 160 mg).

**Table 1 cancers-17-03316-t001:** Baseline demographic and clinical characteristics of patients included in the ReTrITA sub-study.

	ReDOS N (%)	No-ReDOS N (%)	*p*-Value
Total	313 (100)	400 (100)	**0.0011**
Age Median (min–max)	58 (30–86)	59 (33–86)	0.9263
Age ≥70 yrs <70 yrs	137 (43.7) 176 (56.3)	151 (37.8) 249 (62.3)	0.1042
Sex Female Male	131 (41.8) 182 (58.2)	157 (39.3) 243 (60.8)	0.4824
RAS status Wild type Mutant type Unknown	130 (41.5) 175 (56.0) 8 (2.5)	153 (38.3) 235 (58.8) 12 (3.0)	0.6549
Primary tumour location Right side Left side Rectum	95 (30.3) 152 (48.6) 66 (21.1)	141 (35.3) 193 (48.3) 66 (16.5)	0.1948
MMR dMMR pMMR Unknown	11 (3.5) 244 (78.0) 58 (18.5)	7 (1.8) 151 (37.8) 242 (60.5)	**<0.0001**
PS ECOG 0 1 2	117 (37.4) 177 (56.5) 19 (6.1)	113 (28.3) 236 (59.0) 51 (12.8)	**0.0017**
Prior adjuvant therapy Yes No	94 (30.0) 219 (70.0)	97 (24.3) 303 (75.8)	0.0838
Metastatic disease sites Liver only Liver + other Others	55 (17.6) 151 (48.2) 107 (34.2)	36 (9.0) 227 56.8() 137 (34.3)	**0.0019**
CT 1° line regimen Monochemotherapy Doublet chemotherapy Triplet chemotherapy Unknown	18 (5.7) 254 (81.2) 40 (12.8) 1 (0.3)	12 (3.0) 249 (62.3) 15 (3.8) 124 (31.0)	**<0.0001**
CT 2° line regimen Monochemotherapy Doublet chemotherapy Triplet chemotherapy Unknown	29 (9.3) 231 (73.8) 12 (3.8) 40 (13.1)	21 (5.3) 234 (58.5) 4 (1.0) 141 (35.3)	**<0.0001**
Rechallenge therapy yes no	62 (19.9) 251 (80.1)	64 (16.0) 336 (84.0)	0.1861
Biological agents 1° line Anti-EGFR use Anti-VEGF use None	99 (31.6) 160 (51.1) 54 (17.2)	80 (20.0) 160 (40.0) 160 (40.0)	**<0.0001**
Biological agents 2° line Anti-EGFR use Anti-VEGF use None	22 (7.1) 207 (66.1) 84 (26.8)	15 (3.8) 180 (45.0) 205 (51.3)	**<0.0001**

**Table 2 cancers-17-03316-t002:** Efficacy outcomes in patients treated with regorafenib (ReDOS vs. no-ReDOS).

		Redos	No-Redos
**OS**	mOS (months)	7.4	6.7
	3y-OS (%)	2.5	7.2
	2y-OS (%)	8.9	14.0
	HR (95% CI)	1.00 (0.85–1.18)	
	*p*-Value	0.9298	
**PFS**	mPFS (months)	3.1	3.9
	1y-OS (%)	3.8	5.7
	2y-OS (%)	0.6	2.2
	HR (95% CI)	0.76 (0.64–0.89)	
	*p*-Value	**0.0007**	
**ORR**	PR + CR (%)	2.0	2.6
	*p*-Value	0.4386	
**DCR**	PR + CR + SD (%)	23.2	25.3
	*p*-Value	0.6580	
**DoR**	months	8.9	15.4
	HR (95% CI)	1.68 (0.35–7.94)	
	*p*-Value	0.5944	

**Table 3 cancers-17-03316-t003:** Grade 3/4 adverse events in patients treated with regorafenib according to ReDOS versus no-ReDOS dosing strategies in the ReTrITA sub-analysis.

	ReDOS		No-ReDOS		
	*n*	%	*n*	%	*p*-Value
All events G3/G4	141	100	203	100	**0.0008**
All pts who experienced G3/G4 toxicities	111	35.4	158	39.5	**0.0042**
All Haematologic events G3/G4	11	7.8	21	10.3	0.0771
All non Haematologic events G3/G4	130	92.2	182	89.7	**0.0032**
Most common Haematologic toxicities G3/G4					0.9148
Neutropenia	2	18.2	5	23.8	
Febrile neutropenia	0	0.0	0	0.0	
Thrombocytopenia	2	18.2	3	14.3	
Anaemia	7	63.6	13	61.9	
Most common non Haematologic toxicities G3/G4					**0.0157**
Fatigue	46	39.0	48	23.6	
Hand-foot skin reaction	20	16.9	43	21.2	
Hypertension	15	12.7	15	7.4	
Liver dysfunctions	4	3.4	9	4.4	
Diarrhoea	7	5.9	19	9.4	
Skin disorders	2	1.7	14	6.9	
Others	24	20.3	55	27.1	

**Table 4 cancers-17-03316-t004:** Comparison of survival outcomes across the ReTrITA sub-analysis and selected real-world studies of regorafenib in refractory metastatic colorectal cancer (mCRC).

Study	Country/Setting	N Patients	Flexible/Reduced Starting Dose (%)	Median OS (Months)	Median PFS (Months)	Key Notes
**ReTrITA Sub-analysis**	Italy (retrospective, multicentre)	729	~40% dose-escalation (ReDOS-like)	~8–9	~3.0	Improved PFS with escalation; OS comparable; lower G3–4 AEs with escalation
**RE-SEARCH (Muñoz et al. 2025) [32]**	Spain (multicentre, retrospective)	242	63.8% started <160 mg	Not reported	3.0	Escalation increasingly used post-2019; no efficacy differences across dose groups; higher G3–4 AEs with 160 mg start + reductions (40.6%)
**Nakashima (Japan) [33]**	Japan (retrospective, multicentre)	2530	Body-weight–adjusted regimens common	7.5–8.0	~3.0–3.5	Cumulative dose predicted OS; lighter patients tolerated ≤120 mg
**Yan et al. (China) [34]**	China (retrospective)	77	Majority <160 mg	17.8	4.6	Included earlier lines and some IO combinations; superior OS vs. Western cohorts
**Bekaii-Saab et al., 2024 [35]**	USA (claims database)	703	Flexible dosing ↑ from 21% to 45% post-ReDOS guideline	Not primary endpoint	Not primary endpoint	Flexible dosing improved persistence (≥3 cycles: 45% vs. 36% pre-ReDOS)
**CORRELATE (Taiwan, Yeh et al. 2021) [36]**	Taiwan (prospective observational)	128	71.9% started < 160 mg	11.6	2.2	Higher mutation prevalence; OS longer; frequent early dose reductions
**Hatori et al., 2021 [37]**	Japan (retrospective, multicentre)	176	Most started at ≤120 mg	~8.0	~3.0	Association between lower starting dose and preserved efficacy; real-world validation of flexible dosing
**Peeters et al., 2024 [38]**	Europe (France, Italy, Belgium)	355	Flexible regimens: ReDOS-like or dose-adjusted	Not reported	Not reported	Flexible dosing improved duration on regorafenib therapy before discontinuation (≥3 cycles in 67–93% vs. 64% with standard)

## Data Availability

The data to support the results reported in this study are available from the corresponding author on reasonable request.

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
