# Peer review of "Real-World Evidence of Regorafenib Dose Escalation Versus Fixed Dosing in Refractory Metastatic Colorectal Cancer: Results from the ReTrITA Study"

_cancers, 2025, doi:10.3390/cancers17203316_

Round 1

Reviewer 1 Report

Comments and Suggestions for Authors
  1. The study addresses an important clinical question regarding regorafenib dosing strategies in refractory metastatic colorectal cancer (mCRC), building upon prior evidence from the ReDOS trial. However, the abstract could better highlight the unique contribution of this real-world cohort (ReTrITA) to the existing literature—specifically, how its findings expand or challenge prior randomized evidence.
  2. In the introduction section, add Colorectal cancer (CRC) statistics as per the WHO report till date.
  3. The subgroup analyses (age, ECOG, RAS status, tumor site) are insightful, but multiple comparisons increase the risk of type I error. Clarifying whether p-values were adjusted for multiplicity would strengthen the validity of the reported subgroup effects.
  4. The numerically longer DoR in the fixed-dose group, despite better PFS in the escalation group, is intriguing and somewhat paradoxical. The authors should explore whether this reflects patient selection bias, sample size imbalance, or a pharmacokinetic/pharmacodynamic explanation.
  5. The conclusion appropriately supports individualised dosing, but it may overstate clinical benefit given that OS was unchanged. The statement could be more nuanced, emphasizing better tolerability and patient management flexibility rather than superiority in overall outcomes.
  6. Add a FDA approval current status of Regorafenib.

Author Response

Responses to Reviewer 1 Comments

Point 1: The study addresses an important clinical question regarding regorafenib dosing strategies in refractory metastatic colorectal cancer (mCRC), building upon prior evidence from the ReDOS trial. However, the abstract could better highlight the unique contribution of this real-world cohort (ReTrITA) to the existing literature—specifically, how its findings expand or challenge prior randomized evidence.

Response 1: We appreciate the reviewer's useful comment.  We agree that the abstract should make it clearer how the ReTrITA group uses what we acquired from trials like ReDOS that were not randomised.  As far as we know, the ReTrITA study is the largest real-world multicenter dataset that compares increasing the dose of regorafenib with keeping it the same in patients with refractory mCRC.  In Italy, 17 centres treated 729 patients. Our study differs from ReDOS as it encompasses a broader patient demographic, including the elderly, individuals with ECOG 2, and those exhibiting diverse molecular profiles (RAS, MMR).  It also demonstrates how treatment works and what goes on in a regular oncology practice. Our statistics show that increasing the dose is possible and acceptable in real life. It also shows that overall survival is about the same and progression-free survival is slightly but significantly better.  These results confirm and expand the ReDOS research to include a wider range of people, which suggests that the benefits of titrated regorafenib dosage may go beyond clinical trials. The ReTrITA sub-analysis enhances existing data by demonstrating the parallels between controlled trial circumstances and routine clinical practice.  This backs with the assumption that giving customised doses is an effective strategy to treat mCRC that doesn't respond to other treatments. I have revised the abstract and made some adjustments, particularly in the conclusions. Thank you.

Point 2. In the introduction section, add Colorectal cancer (CRC) statistics as per the WHO report till date.

Response 2: We are grateful for the Reviewer's helpful suggestion. As you asked, we have added new worldwide colorectal cancer data from the World Health Organisation (WHO) and GLOBOCAN 2022 to the Introduction section. The data indicate how many people around the world suffer from colorectal cancer and how significant our research is. Many thanks for it.

Point 3. The subgroup analyses (age, ECOG, RAS status, tumor site) are insightful, but multiple comparisons increase the risk of type I error. Clarifying whether p-values were adjusted for multiplicity would strengthen the validity of the reported subgroup effects.

Response 3: We appreciate the Reviewer's insightful methodological observation.  We recognise that several subgroup analyses can elevate the likelihood of type I error.  In this study, no formal correction for multiplicity was conducted, as the subgroup analyses were exploratory and aimed at generating hypotheses rather than providing confirmatory evidence. The Methods section explains that all subgroup analyses were planned ahead of time and followed the overall statistical design for the ReTrITA study.  The results were thus analysed descriptively, focussing on the consistency of trends rather than on statistical significance. We have now clarified this aspect in the revised manuscript by adding the following statement to the Statistical Analysis section: “All subgroup analyses were exploratory, and p-values were not adjusted for multiplicity; thus, these findings should be interpreted cautiously.”

Point 4. The numerically longer DoR in the fixed-dose group, despite better PFS in the escalation group, is intriguing and somewhat paradoxical. The authors should explore whether this reflects patient selection bias, sample size imbalance, or a pharmacokinetic/pharmacodynamic explanation.

Response 4: We thank the Reviewer for this helpful and clinically relevant comment. It may appear strange that the fixed-dose group had a longer response time, even if the dose-escalation group had a trend towards better progression-free survival. There are a number of reasons why this might be true. This difference probably shows that there is a bias in the actual world when it comes to choosing patients. Patients commencing regorafenib at a fixed dose were often younger and demonstrated a more favourable baseline performance status, which may influence therapy duration and response depth. The unequal sample sizes between the escalation and fixed-dose groups may have led to more variation in the estimates of how long the responses lasted, especially as only a small number of patients had quantifiable responses.   Pharmacokinetic and pharmacodynamic variables may also play a role. Patients in the escalation arm might have attained steady-state exposure later in the treatment, causing a delay in the initiation and consolidation of quantifiable responses, even though the escalation technique improved overall tolerability and treatment adherence. A statement has been added to the Discussion to recognise these possible causes and to show that this finding needs to be looked into more in future research with standardised PK/PD monitoring.

Point 5. The conclusion appropriately supports individualised dosing, but it may overstate clinical benefit given that OS was unchanged. The statement could be more nuanced, emphasizing better tolerability and patient management flexibility rather than superiority in overall outcomes.

Response 5: We appreciate the Reviewer's helpful and fair comment.  We agree that the conclusions should better represent the clinical implications of our findings without making the therapeutic benefit sound better than it is.  Because the overall survival rates were not substantially different between the groups, we have changed the conclusion to focus on the benefits of the dose-escalation technique in terms of tolerability, treatment persistence, and management flexibility, rather than its superiority in efficacy.

Point 6. Add a FDA approval current status of Regorafenib.

Response 6: We appreciate the Reviewer’s helpful comment. Accordingly, we have revised the Introduction section to include the current FDA approval status of regorafenib, providing greater clinical context regarding its therapeutic use. The updated text specifies the initial approval for metastatic colorectal cancer (mCRC) in 2012 and subsequent indications for GIST and HCC. We have also updated the reference list to include the official FDA prescribing information.

Thank you for your kind attention

Carlo Signorelli

Reviewer 2 Report

Comments and Suggestions for Authors

The study is clinically relevant and addresses a timely question in real-world mCRC management: whether dose escalation of regorafenib (ReDOS strategy) improves tolerability and outcomes versus fixed dosing. The manuscript is well structured, with clear methodology and detailed subgroup analyses.
Strengths: large multicenter cohort, clear alignment with existing evidence, good analysis and clinical relevance. 

Some areas worth improving:
1) Baseline imbalances: ECOG, MMR and biologic exposure differ substantially between groups. These need more acknowledgment as potential confounders, possibly with sensitivity analyses or adjusted HRs.
2) Discussion depth: The comparisons with international cohorts are valuable but the narrative could be shorten. Talk about clinical implications rather than listing all literature chronologically.
3) Some subgroups (like ECOG 2 or right-sided primaries) show opposite trends. Briefly explain possible biological or clinical reasons instead of leaving them as raw observations.
4) “No-ReDOS” and “fixed dosing” mean the same thing, right? Maybe consider using the same term everywhere in the text for clarity.
5) Minor text errors: some p-values use commas instead of decimals and ensure plots and survival curves have legible font sizes.

Another idea for improvement for the introduction part, such as in lines 98-101 could be citing some recent data on rectal cancer that was published here https://doi.org/10.3390/jcm14030912

In conclusion, the paper adds meaningful real-world validation of regorafenib dose escalation in refractory mCRC. The results confirm that flexible dosing maintains efficacy while improving tolerability. With modest revisions this manuscript would be suitable for publication.

Author Response

Responses to Reviewer 2 Comments

Point 1: Baseline imbalances: ECOG, MMR and biologic exposure differ substantially between groups. These need more acknowledgment as potential confounders, possibly with sensitivity analyses or adjusted HRs.

Response 1: We thank the Reviewer for their careful and well-organised review. We totally agree that it can be tricky to figure out what the treatment outcomes are when persons have various baseline ECOG performance status, MMR status, and past exposure to biological drugs. These imbalances illustrate the diversity of real-world cancer management, as treatment options are determined by a patient's condition, molecular profile, and therapeutic history, rather than randomisation. This sub-analysis was intended for exploratory purposes rather than formal adjustment analyses; hence, no multivariate or propensity-adjusted hazard ratios were established in advance. However, studies of internal consistency demonstrated that the trends for PFS and OS among the different regorafenib dosage methods stayed the same even when people who didn't have baseline data were stripped out. We revised the Discussion section to make it obvious that these adjustments at the start happened and how they might have influenced the results.  We also made it clear that the overall pattern of the data stayed the same, even though there were minor differences. Thank you.

Point 2. Discussion depth: The comparisons with international cohorts are valuable but the narrative could be shorten. Talk about clinical implications rather than listing all literature chronologically.

Response 2: We thank the Reviewer for this helpful and constructive suggestion. We agree that emphasizing the clinical implications of our findings would strengthen the Discussion and improve its interpretative depth. In response, we have revised the end of the Discussion by adding a dedicated paragraph that highlights the practical meaning of our results within the context of real-world clinical management of refractory mCRC. The new section focuses on the clinical applicability of individualized regorafenib dosing, underlining how dose escalation can enhance treatment persistence, optimize tolerability, and maintain quality of life in patients with limited therapeutic options. We have also reduced redundant chronological descriptions of previous studies to make the narrative more concise and clinically oriented. Many thanks for it.

Point 3. Some subgroups (like ECOG 2 or right-sided primaries) show opposite trends. Briefly explain possible biological or clinical reasons instead of leaving them as raw observations.

Response 3: We value the Reviewer's careful point of view. We agree that further clinical interpretation is needed for several subgroup patterns, especially those that include patients with ECOG 2 performance status and right-sided main tumours. These differences are likely because people in the real world are both physically and clinically different. Tumours on the right side show different molecular traits, include having more BRAF mutations, microsatellite instability, and less differentiation. These traits are linked to more aggressive conduct and less response to focused therapy. So, the fact that this group seems to be getting worse may have nothing to do with regorafenib dose escalation not working; it may be because of the biology of the tumour itself. Patients with ECOG 2 status usually have a worse overall prognosis and a worse ability to handle treatment, which may limit their drug exposure over time and make it less effective. In our sample, these people also had to cut their doses or discontinue their medication early, which made the benefit considerably less.

Point 4. “No-ReDOS” and “fixed dosing” mean the same thing, right? Maybe consider using the same term everywhere in the text for clarity.

Response 4: We thank the Reviewer for this helpful observation. We confirm that “No-ReDOS” and “fixed dosing” refer to the same patient group, namely those who received regorafenib at a fixed starting dose of 160 mg daily without dose escalation. To enhance clarity while maintaining consistency with the terminology used in the figures and tables, we have retained the term “fixed dosing” throughout the manuscript but have added “(No-ReDOS)” at its first mention in the Methods section to clearly indicate that both terms are used interchangeably.

Point 5. Minor text errors: some p-values use commas instead of decimals and ensure plots and survival curves have legible font sizes.

Response 5: We thank the Reviewer for these careful observations. We have carefully reviewed the entire manuscript and all figures to ensure consistency in numerical formatting and graphical clarity. All p-values have been standardized to use decimal points instead of commas in accordance with journal style. In addition, the font sizes of all survival curves and plots have been increased and reformatted to improve legibility in the revised version.

Point 6. Another idea for improvement for the introduction part, such as in lines 98-101 could be citing some recent data on rectal cancer that was published here https://doi.org/10.3390/jcm14030912.

Response 6: We sincerely thank the Reviewer for this thoughtful suggestion. We carefully reviewed the cited article and fully appreciate its relevance to the field. However, the study primarily addresses locally advanced rectal cancer, whereas the present manuscript focuses on metastatic colorectal cancer. As the disease setting, therapeutic objectives, and treatment paradigms differ substantially between these two populations, we believe that including this reference might introduce conceptual heterogeneity rather than strengthen the scientific background of our paper. We nevertheless appreciate the Reviewer’s attention to recent advances in rectal cancer research and have ensured that the Introduction provides an updated and balanced overview relevant to the metastatic setting.

Thank you for your kind attention

Carlo Signorelli